# Comparing estimates of psychological distress using 7-day and 30-day recall periods: Does it make a difference?

**Miranda R. Chilver**[1]*, **Richard A. Burns**[1], **Ferdi Botha**[2,3], **Peter Butterworth**[1,2]

**1** National Centre for Epidemiology and Population Health, The Australian National University, Canberra, Australian Capital Territory, Australia, **2** Melbourne Institute: Applied Economic and Social Research, The University of Melbourne, Melbourne, Victoria, Australia, **3** ARC Centre of Excellence for Children and Families Over the Life Course, The University of Melbourne, Melbourne, Victoria, Australia

* miranda.chilver@anu.edu.au

**Data Availability Statement:** All data, analysis script, and outpus are available from the Open Science Framework (url: https://osf.io/aezt6/, DOI: 10.17605/OSF.IO/AEZT6).

## Abstract

Self-report measures are widely used in mental health research and may use different recall periods depending on the purpose of the assessment. A range of studies aiming to monitor changes in mental health over the course of the COVID-19 pandemic opted to shorten recall periods to increase sensitivity to change over time compared to standard, longer recall periods. However, many of these studies lack pre-pandemic data using the same recall period and may rely on pre-existing data using standard recall periods as a reference point for assessing the impact of the pandemic on mental health. The aim of this study was to assess whether comparing scores on the same questionnaire with a different recall period is valid. A nationally representative sample of 327 participants in Australia completed a 7-day and 30-day version of the six-item Kessler Psychological Distress Scale (K6) and a single-item measure of psychological distress (TTPN item) developed for the Taking the Pulse of the Nation survey. Linear mixed models and mixed logistic regression models were used to assess whether altering the recall period systematically changed response patterns within subjects. No substantive recall period effects were found for the K6 or the TTPN, although there was a trend towards higher K6 scores when asked about the past 30 days compared to the past 7 days ($b = 1.00$, 95% CI: -0.18, 2.17). This may have been driven by the "feeling nervous" item which was rated higher using the 30-day compared to the 7-day recall period. Neither the K6 nor the TTPN item were significantly affected by the recall period when reduced to a binary variable of likely severe mental illness. The results indicate that altering the recall period of psychological distress measures does not substantively alter the score distribution in the general population of Australian adults.

## Introduction

Self-report measures are ubiquitous in mental health research as they provide convenient subjective insight to a person's current psychological state. Psychological self-report measures ask

**Funding:** F.B. was supported by the Australian Government through the Australian Research Council's Centre of Excellence for Children and Families over the Life Course (Project ID: CE200100025). The funder had no role in study design, data collection and analysis, decision to publish, or preparation of the manuscript. URL: arc.goc.au M.R.C., R.A.B., and P.B. received no specific funding for this work.

**Competing interests:** The authors MRC, RAB, FB, and PB have declared that no competing interests exist.

participants to report the symptoms they have experienced over a specified timeframe. These reporting timeframes or recall periods differ between self-report scales, and different recall periods might be used with the same scale depending on the purpose of the assessment. For instance, the Kessler Psychological Distress Scale was designed using a 30-day recall period [1] but was adapted to a 12-month recall period to make it comparable to other scales included in the US National Health Interview Survey [2] and can be shortened to a 7-day recall period [3]. Similarly, the SF-36 health measure has been adapted from the original 30-day recall period to a 7-day recall period [4]. Longer recall periods are better suited to capturing less frequent or more stable phenomenon while shorter recall periods are better suited to more frequent or less stable phenomenon [5]. Using a shorter recall period might also reduce prevalence estimates as there is less opportunity to have experienced the condition of interest in a shorter timeframe relative to a longer timeframe [6]. However, the correspondence between prevalence and recall period may differ for scale scores that ask about the frequency of symptoms rather than the presence or intensity of symptoms [7].

The COVID-19 pandemic introduced a period of rapid change and possible instability in mental health in the general population. Because short recall periods are better suited to capturing changes in mental health than longer recall periods [4], many studies monitoring the mental health impact of the pandemic opted to use scales with shorter recall periods of one or two weeks [8–11]. One example of this is the Australian Taking the Pulse of the Nation (TTPN) survey which measured psychological distress over 50 waves during the COVID-19 pandemic using a 7-day recall period instead of the standard 30-day recall period applied in comparable psychological distress measures [9]. However, pre-existing indicators of population level psychological distress using the K6 that could provide a baseline indicator of distress were collected using a 30-day recall period. The aim of the current study was to assess whether changing the recall period from 30 days to 7 days systematically alters psychological distress scores, specifically the widely used 6-item Kessler Psychological Distress Scale (K6) and the single psychological distress item applied in the TTPN survey. This information would be useful in determining whether psychological distress ratings obtained using a 7-day recall period, such as in the TTPN survey, could be compared to pre-existing data that used a 30-day recall period.

Self-report measures requiring participants to recall past events are vulnerable to inaccuracies and biases in memory recall [12, 13]. Shorter recall periods tend to provide more accurate reports than longer recall periods because they enquire about more recent events that are remembered more easily. Thomas and Diener [13] compared participants' daily emotional ratings taken over a period of three or six weeks to their retrospective emotional ratings over the full period. They found that, compared to daily reports, participants tended to overestimate the intensity of both positive and negative emotions in retrospective reports and to overestimate the frequency of their negative emotions relative to positive emotions. A more recent study similarly reported that participants overestimated their negative emotions in retrospective reports compared to daily reports, though this may be specific to depression and anxiety symptoms [12]. Further investigation suggested that the overestimation of negative emotions was partly explained by participants' most intense emotions during the recall period. As the likelihood of more intense emotions is greater over longer recall periods, this could explain why some studies have found that 30-day recall periods result in higher estimates of negative emotions than 7-day recall periods [14]. Though changing the recall period has been found in some studies to change average scores, some studies examining self-reported physical symptoms including fatigue and urinary tract symptoms have found no difference between using 7-day or 4-week recall periods [15, 16].

Varying the recall period could also affect responses by changing how participants interpret the survey questions [7, 17]. Given the brevity of self-report items, participants might look to

contextual information such as the recall period to decide what type of information is relevant to the question. Participants may assume a question using shorter recall periods is focused on more frequent, less intense experiences, whereas longer recall periods imply the question is asking about less frequent, more intense experiences [17]. This can result in participants under-reporting frequencies when asked about longer timeframes compared to shorter timeframes as more frequent events may be viewed as irrelevant to the question.

Although the prevalence of mental disorders is directly related to the reporting timeframe, this may not be the case for self-report measures based on symptom frequency instead of incidence, such as the K6. Specifically, the K6 uses a scale ranging from "none of the time" to "all of the time." Events that never occurred in the past 7 days might have occurred in the past 30 days, increasing the minimum scores. However, events that occurred in the past 7 days did not necessarily occur more frequently over the past 30 days, and in some cases may even have been experienced less frequently when averaged over the longer timeframe. This effect should limit the degree to which total scale scores increase from the 7-day to 30-day recall period, and most of the change would be expected among individuals selecting "none of the time" on the 7-day version.

Given the importance of identifying appropriate pre-COVID mental health measures, understanding the impact of the recall period on estimates of distress and other psychological health outcomes has new prominence. To provide clarity about whether prevalence of psychological distress can be compared between pre-pandemic measures using a 30-day recall period and post-pandemic measures using a 7-day recall period, this study investigates whether changing the recall period on the K6 and TTPN item changes the distribution of responses. Previous reports comparing 30-day and 7-day recall periods on different self-report measures have shown mixed results, with some measures showing no change in the distribution whereas others show a higher central tendency for 30-day relative to 7-day recall periods [4, 14]. However, if any effect of the recall period exists, it was expected that this would be in the direction of higher scores for longer recall periods.

## Materials and methods

### Participants

A general population sample was recruited through the Australian Online Research Unit (ORU) panel during the first two weeks of December 2021. The ORU is an online survey platform that uses both online and offline recruitment methods to build a regionally representative panel from the general public interested in contributing to research. The ORU randomly selects potential participants who meet the eligibility criteria and provides an incentive for their participation. The authors did not have access to identifying information about the participants at any point during the study. This study sought adults 18 years of age and older currently residing in Australia with an even split between males and females, and approximately equal representation across age groups. This study did not apply exclusion criteria. Based on a power analysis, a sample of at least 250 participants was sought to have 75% power to detect an effect size of 0.15 on the TTPN single item scale. The study was approved by the Australian National University Human Research Ethics Committee (Protocol 2021/736) and all participants provided written informed consent prior to participation.

### Measures

**Kessler psychological distress scale.** Participants were asked to complete the K10, a widely applied measure of psychological distress often used to identify serious mental illness [1]. From the K10 items, the six items that form the shortened K6 scale were used to evaluate

K6 scores. K6 scores were focus of the current analysis for comparison with the TPPN survey which included the K6 in two waves. These items ask participants how often they felt nervous, hopeless, restless or fidgety, so sad that nothing could cheer them up, that everything was an effort, and worthless. Response options range from *none of the time* to *all of the time* on a 5-point scale. Participants were asked to consider the past 7 days and the past 30 days in providing their responses. Scores on the K6 range from 6–30, with scores of 19 or higher indicating high levels of psychological distress and likely severe mental illness [3].

**TTPN psychological distress item.**   A single-item adaptation of the K6 was developed for use in the TTPN survey as an ultra-short measure of psychological distress [9]. This item asks participants, "*During the past [recall period], about how often did you feel depressed or anxious?*" with the same response options as the K6. Scores on this measure range from 1 to 5, with scores of 4 or 5 indicating high levels of psychological distress and likely severe mental illness.

The ORU questionnaire also included a measure of age group and sex to be included as covariates in the analysis.

## Procedure

All participants completed the K6 and TTPN psychological distress item with both the 7-day and 30-day recall period. The order of the K6 and TTPN and recall period order was counterbalanced across participants. After completing the K6 and TTPN item (with either the 7-day or 30-day recall period), participants completed an unrelated distractor task before completing the K6 and TTPN item again with the alternate recall period. The entire survey took 10 minutes to complete on average, with the interim distractor task taking approximately 7 minutes.

**Statistical analysis.**   Linear mixed models correcting for statistical non-independence due to repeated measures were used to test whether mean scores on the K6 and TTPN mental distress item differed according to the recall period. Age and gender were included as covariates in the model. Furthermore, to test whether any potential differences between the scale would be perceived as clinically relevant, a mixed logistic regression model was used to assess whether the recall period would influence the probability of falling into the likely severe mental illness category from either measure.

## Results

All study data and results are provided in a data repository on the Open Science Framework (https://osf.io/aezt6/). The responding sample consisted of 330 participants, three of whom were excluded due to missing data. The age and gender distribution of the remaining 327 respondents are provided in Table 1. There was approximately equal distribution across age

**Table 1. Participant characteristics.**

| Characteristic | N | % |
|---|---|---|
| Age group | | |
| 18–24 | 59 | 18.0 |
| 25–34 | 57 | 17.4 |
| 35–44 | 68 | 20.8 |
| 45–54 | 40 | 12.2 |
| 55–64 | 63 | 19.3 |
| 65+ | 40 | 12.2 |
| Gender | | |
| Male | 166 | 50.8 |
| Female | 161 | 49.2 |

**Table 2. Mean and standard deviation of psychological distress and likely severe mental illness.**

| Scale/item | | 7-day | 30-day |
|---|---|---|---|
| | | M (SD) | M (SD) |
| K6 Total | | 12.15 (5.79) | 12.41 (5.79) |
| | 1. Nervous | 2.20 (1.07) | 2.30 (1.09) |
| | 2. Helpless | 1.92 (1.10) | 1.94 (1.11) |
| | 3. Restless or fidgety | 2.13 (1.07) | 2.11 (1.09) |
| | 4. Like everything is an effort | 2.21 (1.15) | 2.25 (1.15) |
| | 5. So sad nothing could cheer up | 1.81 (1.12) | 1.87 (1.11) |
| | 6. Worthless | 1.88 (1.11) | 1.94 (1.15) |
| Severe Mental Illness | | N = 48 (14.7%) | N = 51 (15.6%) |
| TTPN Psychological Distress Measure | | 2.27 (1.11) | 2.26 (1.06) |
| Severe Mental Illness | | N = 48 (14.7%) | N = 46 (14.1%) |

groups, though with fewer participants aged 45–54 and over 65 compared to other age ranges, and approximately equal numbers of men and women. Mean scores on the K6 and TTPN psychological distress scale by recall period are provided in Table 2. Means and standard deviations for individual items of the K6 and TTPN were similar regardless of the recall period. The prevalence of likely severe mental illness was similar for the K6 and TTPN and both scales showed similar numbers of participants falling into the category regardless of the recall period.

*K6*. Linear mixed models (Table 3) showed no effect of the recall period on mean K6 scores, although there was a trend towards higher reported frequencies using the 30-day recall period compared to the 7-day recall period (95% CI: -0.18, 2.17). This effect is illustrated in Fig 1. There was no evidence of either a main effect of time (95% CI: -1.31, 1.05) or an interaction between time and recall period (95% CI: -3.76, 0.85). The random effects logistic regression model did not indicate an effect of the recall period on whether participants fell into the category of likely significant mental illness (OR = 1.20, *p* = 0.566, 95% CI: 0.64, 2.50). However, consistent with the overall score, there was a slight tendency towards more psychological distress for the 30-day recall period compared to the 7-day recall period. There was no evidence of an effect of time (OR 95% CI: 0.43, 1.64) or interaction between time and recall period (OR 95% CI: 0.27, 2.23). The covariates from the K6 models indicated that males had lower distress

**Table 3. Regression model results for total K6 scale and TTPN item.**

| Variable | | K6 Total | | | 95% CI | | TTPN item | | | 95% CI | |
|---|---|---|---|---|---|---|---|---|---|---|---|
| | | B | SE | P | LL | UL | B | SE | P | LL | UL |
| Intercept | | 13.95 | 0.81 | 0.000 | 12.36 | 15.50 | 2.60 | 0.15 | 0.000 | 2.30 | 2.90 |
| 30 day recall | | 1.00 | 0.60 | 0.097 | -0.18 | 2.17 | 0.16 | 0.11 | 0.164 | -0.06 | 0.38 |
| Time 2 | | -0.13 | 0.60 | 0.832 | -1.31 | 1.05 | 0.85 | 0.11 | 0.458 | -0.14 | 0.31 |
| Recall x Time | | -1.46 | 1.18 | 0.215 | -3.76 | 0.85 | -0.34 | 0.22 | 0.119 | -0.77 | 0.087 |
| Male | | -0.42 | 0.59 | 0.475 | -1.57 | 0.73 | -0.19 | 0.11 | 0.085 | -0.40 | 0.03 |
| Age | 25–34 | 1.16 | 0.98 | 0.236 | -0.76 | 3.09 | 0.17 | 0.18 | 0.343 | -0.18 | 0.53 |
| | 35–44 | -0.76 | 0.94 | 0.418 | -2.61 | 1.09 | -0.21 | 0.18 | 0.224 | -0.56 | 0.13 |
| | 45–54 | -2.29 | 1.08 | 0.034 | -4.41 | -0.17 | -0.37 | 0.20 | 0.064 | -0.76 | 0.02 |
| | 55–64 | -3.90 | 0.96 | 0.000 | -5.77 | -2.02 | -0.65 | 0.18 | 0.000 | -1.00 | -0.30 |
| | 65+ | -4.39 | 1.08 | 0.000 | -6.51 | -2.26 | -0.72 | 0.20 | 0.000 | -1.11 | -0.32 |

*Notes*. Reference age group is 18–24 years.

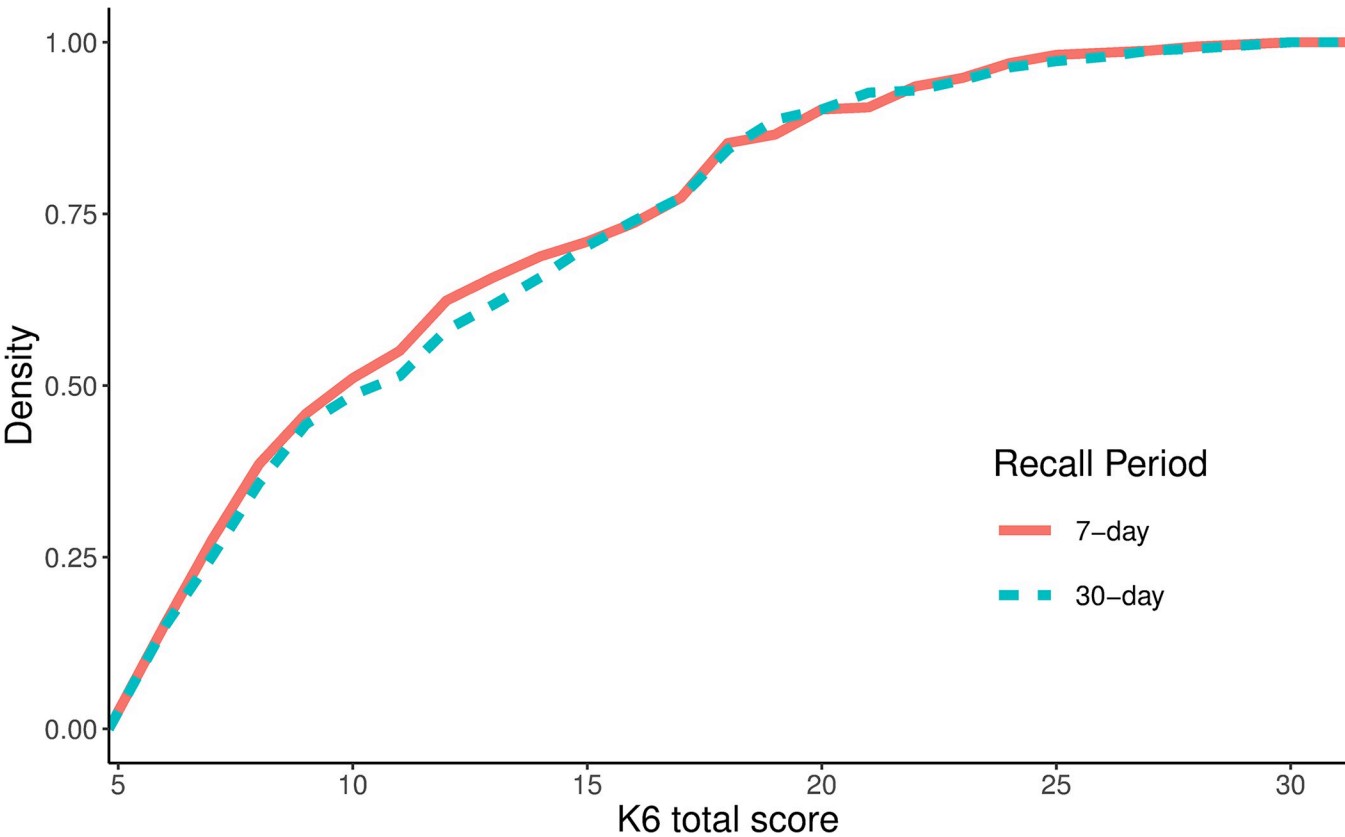

**Fig 1. Cumulative distribution of the K6 total score using a 7-day versus 30-day recall period.**

levels than females (though not significantly) and that older adults generally reported less distress than younger respondents.

Further investigation of each of the individual K6 items found that mean scores on the item *"how frequently did you feel nervous?"* was significantly but not substantively impacted by the recall period whereby participants tended to respond with higher scores when asked about the past 30 days relative to being asked about the past 7 days ($b = 0.09$, $p = 0.024$, 95% CI: 0.01, 0.17). The cumulative distribution plot of this item (Fig 2) shows that the change is largely driven by a shift from "none of the time" to "a little of the time". No other items showed a significant effect of the recall period on mean scores.

## TTPN mental distress item

The linear mixed model results for the TTPN item are shown in Table 3. There was no significant effect of recall period on the mean TTPN item score (95% CI: -0.06, 0.38), no effect of time (95% CI: -0.14, 0.31) and no interaction between time and recall period (95% CI: -0.77, 0.09). The logistic mixed model similarly showed no effect of recall period on whether participants fell into the category of likely significant mental illness (OR = 0.75, $p = 0.595$, 95% CI: 0.26, 2.15). This was also reflected in the cumulative distribution plot shown in Fig 3, which follows the same pattern in both the 7-day and 30-day recall period.

## Discussion

The aim of this study was to investigate whether the score distributions of psychological distress according to the K6 and TTPN item are sensitive to whether a 7-day or 30-day recall

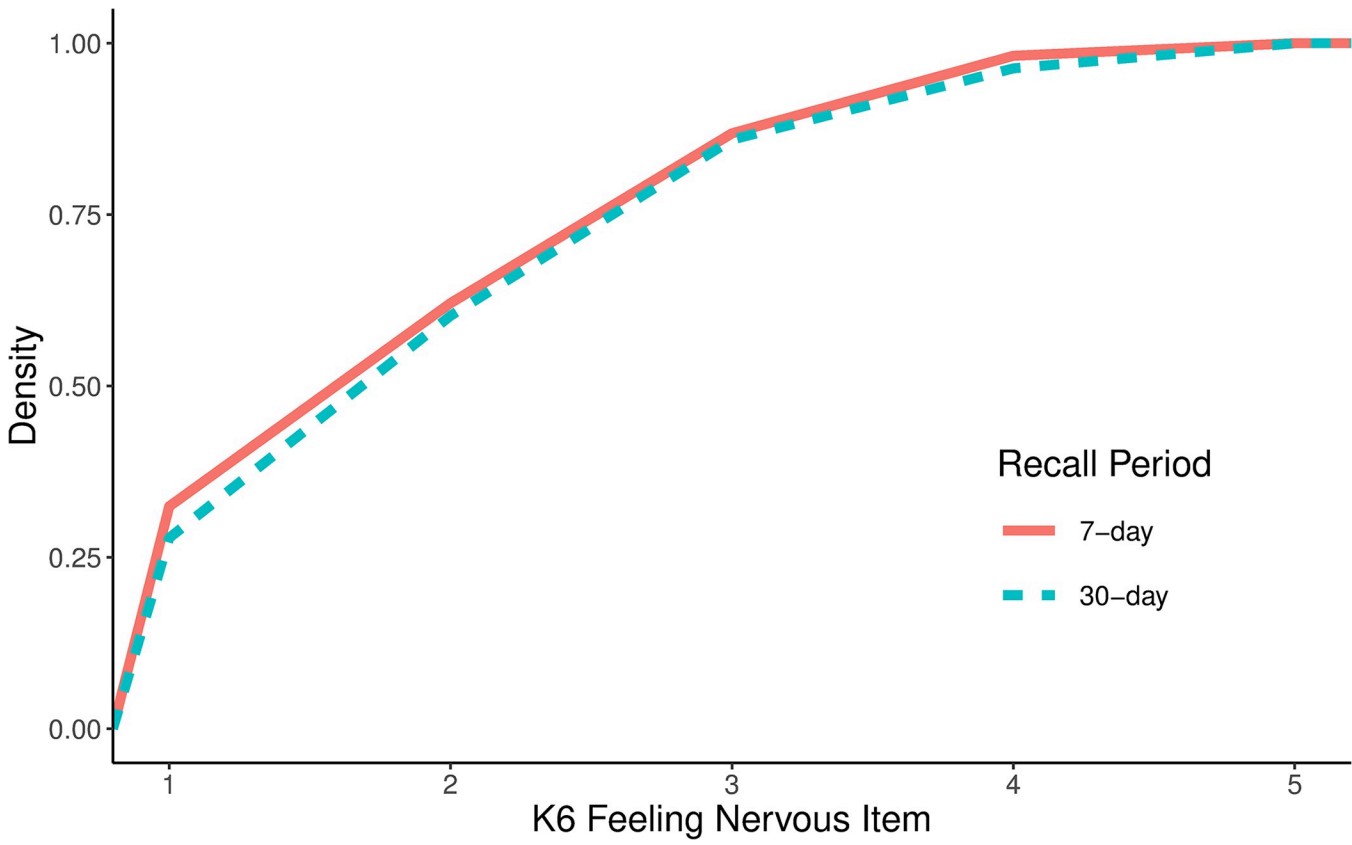

**Fig 2. Cumulative distribution of the K6 "feeling nervous" item using a 7-day versus 30-day recall period.**

period is used. The results indicate that any effect of the recall period on mean scores is very small and there is no evidence that the difference between recall periods is of clinical relevance based on no significant change to the binary outcome of likely severe mental illness on either the K6 or the TTPN item. There was a tendency towards higher mean scores when using a 30-day recall period compared to a 7-day recall period, but effect sizes were consistently small and did not reach statistical significance for the total scores. When considering potential practical implications, comparing estimates of psychological distress during the COVID-19 pandemic based on the 7-day recall period with pre-pandemic measures using a 30-day recall period would (if anything) underestimate the mental health impact of the pandemic. However, the scale differences are small enough that the prevalence of likely severe mental illness can be fairly compared between these recall periods for both the K6 and the TTPN item.

Although there was no overall effect of the recall period on mean scores of either the K6 or the TTPN item, one K6 item asking how often respondents felt nervous did show a small but significant increase in the mean score when comparing the 30-day recall period to the 7-day recall period. This is in line with previous studies comparing 30- and 7-day versions of self-report measures that have found trends towards higher scores over longer recall periods [4, 14]. This effect could reflect a tendency to overestimate the occurrence of emotional states over longer recall periods [12, 13], however, it was not reflected in any other items measuring similar constructs. The cumulative distribution plot shows that the change in the distribution between the 7-day and 30-day version is driven by an upshift of low scorers such that individuals who reported feeling nervous "none of the time" when asked about the past 7 days were

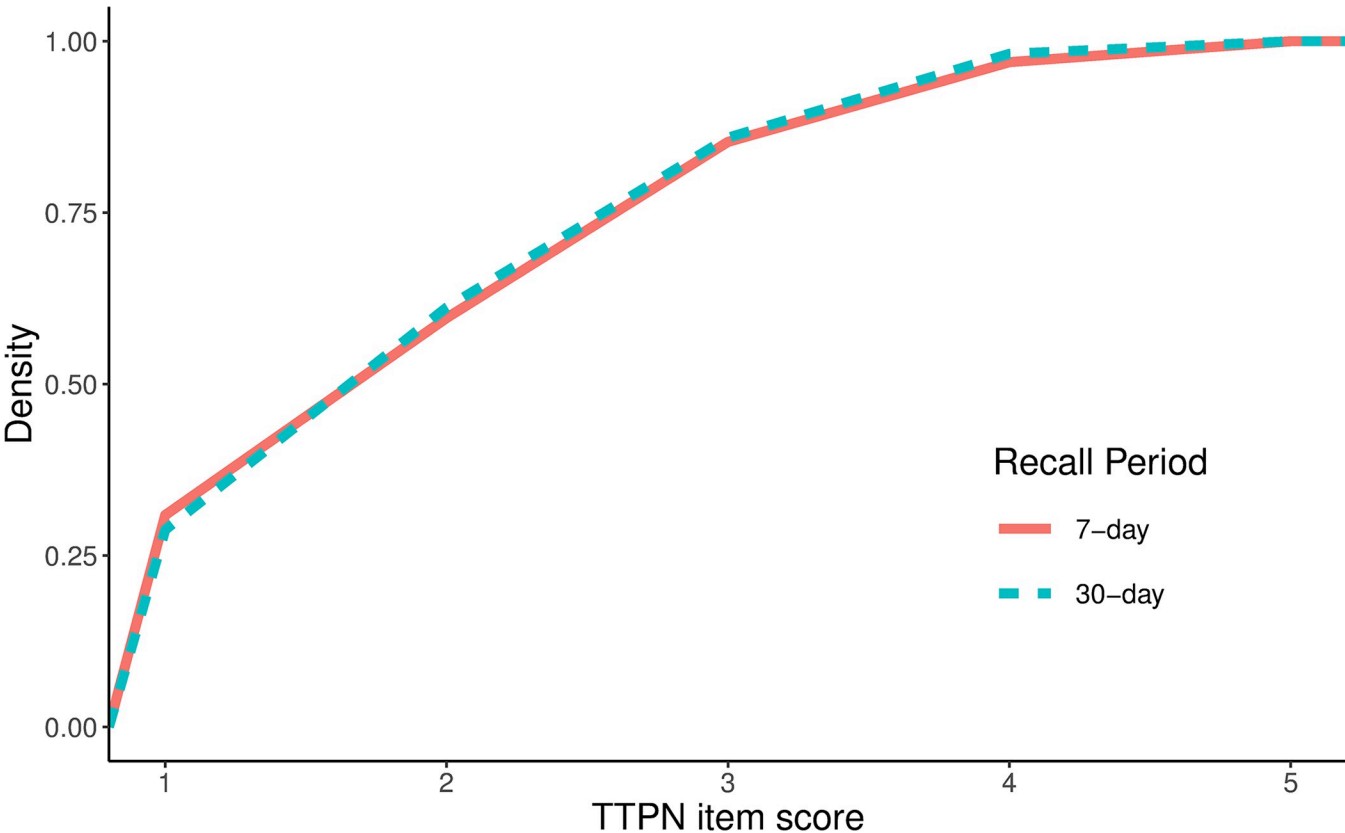

**Fig 3. Cumulative distribution of the TTPN psychological distress item using a 7-day versus 30-day recall period.**

more likely to select "a little of the time" when asked about the past 30 days. This item also had one of the highest mean scores overall, suggesting that feeling nervous is among the more common reported symptoms of psychological distress. As such, it could be that the shift in responses specific to this item simply reflects a higher likelihood of feeling nervous over a month-long period relative to the other symptoms included in the K6. Nevertheless, this shift is not clinically relevant as it predominantly reflects a change in individuals who infrequently experience nervousness.

Previous studies using undergraduate samples have suggested that negative emotions are more susceptible to recall bias than positive emotions [12, 13]. However, the current study used a larger, more representative sample and found that only nervousness was significantly affected by the recall period, and this effect was very small. This suggests that self-reported emotional symptoms are not necessarily sensitive to the recall period. Future research could investigate specific symptoms more closely, for example by comparing multi-item scales that focus on more specific symptoms, such as anxiety symptoms only, to clarify whether certain symptoms are more sensitive to changes in the recall period. The current results found that neither the K6 or the TTPN measures were affected by the recall period, but both cover a combination of depression and anxiety symptoms. Another question is whether participants consider the same types of emotional experiences to be relevant to responding to both the 7-day and 30-day versions of the questionnaires. It has been shown previously that obtaining similar mean scores for different recall periods does not necessarily mean that the participants are considering the same type of events in both recall periods [7]. Finally, while the current study

sought a representative participant sample, some populations may be missed by the ORU panel. For instance, the sample was not stratified by education or household income, and these factors were not controlled for in the current analysis. The comparability of the K6 and TTPN psychological distress measures over the 7- and 30-day recall periods might therefore not generalise across all population groups.

## Conclusions

In sum, this study found that altering the recall period of the K6 and TTPN psychological distress item from 30 days to 7 days does not substantively alter the score distribution in a general population of Australian adults. An implication of this result is that pre-pandemic measures of psychological distress in Australia using the 30-day recall period can be used as a reference point for post-pandemic measures taken using the 7-day period, pending comparability of samples and research methods. For example, if pre-pandemic K6 scores obtained using the 30-day recall period were to be used to indicate baseline psychological distress in the Australian population, this would provide a conservative comparison to post-pandemic measures taken as part of the TTPN survey using a 7-day recall period when using total scores or in estimating the prevalence of severe distress. The current work suggests that the recall period should be selected chosen according to the application, with shorter recall periods being more sensitive to change over time and longer recall periods providing more insight into general symptom levels and prevalence rates [5, 6]. This is also the case when using the established cut-points in a clinical setting. Further research is needed to clarify which emotional and physical symptoms are and are not sensitive to changes in recall periods.

## Acknowledgments

The authors acknowledge the role of the Online Research Unit in participant recruitment and thank Carmel Poyser for assisting in early stages of this project.

## Author Contributions

**Conceptualization:** Richard A. Burns, Peter Butterworth.

**Data curation:** Richard A. Burns.

**Formal analysis:** Miranda R. Chilver, Richard A. Burns, Peter Butterworth.

**Methodology:** Richard A. Burns, Peter Butterworth.

**Project administration:** Richard A. Burns.

**Supervision:** Peter Butterworth.

**Writing – original draft:** Miranda R. Chilver.

**Writing – review & editing:** Miranda R. Chilver, Richard A. Burns, Ferdi Botha, Peter Butterworth.

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
