## [Decision Letter · Decision Letter 0]

6 Sep 2023

PONE-D-23-09923Comparing estimates of psychological distress using 7-day and 30-day recall periods: Does it make a difference?PLOS ONE

Dear Dr. Chilver,

Thank you for submitting your manuscript to PLOS ONE. After careful consideration, we feel that it has merit but does not fully meet PLOS ONE’s publication criteria as it currently stands. Therefore, we invite you to submit a revised version of the manuscript that addresses the points raised during the review process.

We look forward to receiving your revised manuscript.

Kind regards,

Qin Xiang Ng, MBBS, GDMH, MPH

Academic Editor

PLOS ONE

Journal Requirements:

2. We noted in your submission details that a portion of your manuscript may have been presented or published elsewhere:

"The paper has been posted online to the Melbourne Institute Working Papers Series without undergoing peer review (https://melbourneinstitute.unimelb.edu.au/publications/working-papers/search/result?paper=4385776). The working paper series is a form of preprint and not considered as dual publication. This article is not under consideration for publication in any other journal."

Please clarify whether this publication was peer-reviewed and formally published. If this work was previously peer-reviewed and published, in the cover letter please provide the reason that this work does not constitute dual publication and should be included in the current manuscript.

Reviewers' comments:

Reviewer's Responses to Questions

**Comments to the Author**

1. Is the manuscript technically sound, and do the data support the conclusions?

Reviewer #1: Yes

Reviewer #2: Yes

Reviewer #3: Yes

2. Has the statistical analysis been performed appropriately and rigorously?

Reviewer #1: Yes

Reviewer #2: Yes

Reviewer #3: Yes

3. Have the authors made all data underlying the findings in their manuscript fully available?

Reviewer #1: Yes

Reviewer #2: No

Reviewer #3: Yes

4. Is the manuscript presented in an intelligible fashion and written in standard English?

Reviewer #1: Yes

Reviewer #2: Yes

Reviewer #3: Yes

5. Review Comments to the Author

Reviewer #1: This is a well-written manuscript addressing the issue of recall period and whether it matters with respect to 7 versus 30 days in the measurement of psychological distress. The analyses are clearly outlined and presented, and the conclusions are reasonable, generally within the parameters of the study (e.g., generalized to general Australian population).

Introduction: while interesting, it seems a little long and could be tightened by removing some tangents.

pp 3-4 (recall bias section): the authors go into some detail regarding work that suggest recall bias is increased with longer time periods and other factors, but there was no mention of studies that have shown no difference in reports when recall period varies, including 7- versus 30-day recall periods, and there are several such studies. in the end of the introduction, the authors state: "Previous reports comparing 30-day and 7-day recall periods on different self report measures have shown mixed results, with some measures showing no change in the distribution whereas others show a higher central tendency for 30-day relative to 7-day recall periods (4,14). again, no citations for studies that have found no difference. (e.g., K Flynn et al J Urol 2019; Lai et al, J Clin Epi 2009).

Did the authors examine whether there is any effect of order (first versus second) on responses? I appreciate the counterbalancing to control for order effects, but it remains of interest whether the second time around leads to more reporting (regardless of whether the second time around was 7-day or 30-day.

General: The approach to introduction and discussion generally reveals an experimenter mindset that there should be a difference where there was none. I mentioned this in intro, but then again in discussion are comments like: "Future research should investigate why certain emotional states might be more sensitive to changes in the recall period than others" and "caution should be taken in interpreting changes in individual items of the K6 scale as some items are more sensitive to the changed recall period than others". Perhaps a good balance would be to recommend more research to identify emotional (and other) states that are not sensitive to recall period and yet still valid."

Minor:

Intro: "The aim of the current study was to assess whether changing the recall period from 30 days to 7 days systemically alters ..." do you mean "systematically"?

Procedure: " After completing the K6 and TTPN item (with either the 7-day or 30-day recall period), participants completed an unrelated distractor task before completing the K6 and TTPN item again with the alternate recall period ..." Please provide the average amount of time the distractor task took.

Reviewer #2: An extremely clear and well-written piece of work, which answers an interesting question, which is relevant to many researchers who frequently use self-reported mental health scales. The authors investigated the effect that changing the recall period from seven to thirty days for the K6 and TTPN measures had on participant responses. There was a slight trend for higher scores for the thirty-day recall period, although this was not statistically significant and crucially did not change the interpretation of whether someone would be classified as having severe psychological distress. I was glad to see that they considered the effects of individual items from the K6 rather than just the overall sum score. However, I was expecting to see a Table of results for the mixed effects models (the main analysis), could this be added for review prior to publication?

Reviewer #3: This paper investigates a useful question for those interested in survey methodology related to mental health. Differences that might emerge depending on the requested recall period would have significant implications for future data collection and clinical considerations that rely on population surveys. This paper provides useful insights on the relative similarity among both 7-day and 30-day recall periods for self-assessed psychological distress.

Overall, I think the paper is well written, and both the methods and results are robust and accurately interpreted. The findings are straightforward and communicated well.

I do think, however, that more consideration could be given to the limitations of this study in the discussion section. The background section does a very good job highlighting some of the issues that might relate to differences in responses for alternate recall periods, and there is some good reflection of these issues in the discussion. Still, there are some specific limitations that I think deserve greater attention to strengthen the discussion and draw out the implications of the research:

• The methods section describes how the ORU sample was constructed, indicating that age and gender were collected to serve as covariates. I think the discussion should reflect on the limitations from not including other potentially important covariates, particularly those related to education, socioeconomic status, or other physical health measures. Clearly, the sample size restricts the complexity of the model, but I think the paper would benefit from more consideration of this idea.

• Relatedly, I think the paper should include some mention of the results related to the age and gender covariates. Perhaps there were no statistically significant results for these, but their inclusion in the models could also have indicated some kind of stratified differences, such that, for example, a difference between recall periods was identified for a specific age group. Such findings would inform future survey strategies.

• The text might also benefit from considerations of the difference in the K6 and TTPN measures themselves, and what this might mean for other mental health measures used in population surveys. In other words, some surveys include CES-D for depression or GAD-7 for anxiety, which are intended to be more specific than general psychological distress. This would be for future research, but might be a point to briefly highlight in the limitations.

The conclusions around the applicability of 7-day recall period responses to pre-pandemic 30-day recall measures are strong and supported well by the analysis presented. The authors might want to consider including some reflections or recommendations on the pros and cons of using 7-day vs 30-day measures in future survey settings. There might also be value in considering the relevance of these findings for clinical settings related to mental health and psychology, given the logistic regression analyses covering likely clinical poor mental health.

6. PLOS authors have the option to publish the peer review history of their article (what does this mean?). If published, this will include your full peer review and any attached files.

**Do you want your identity to be public for this peer review?** For information about this choice, including consent withdrawal, please see our Privacy Policy.

Reviewer #1: No

Reviewer #2: No

Reviewer #3: No

---

## [Author Response · Author response to Decision Letter 0]

31 Oct 2023

Reviewer #1

This is a well-written manuscript addressing the issue of recall period and whether it matters with respect to 7 versus 30 days in the measurement of psychological distress. The analyses are clearly outlined and presented, and the conclusions are reasonable, generally within the parameters of the study (e.g., generalized to general Australian population).

Point 1. Introduction: while interesting, it seems a little long and could be tightened by removing some tangents. 

Response: We have endeavored tighten the argument and trim the introduction to be more succinct. The word count has now been reduced from 1098 to 1063 words, while also adding additional information to the introduction as requested by reviewers. 

Point 2. pp 3-4 (recall bias section): the authors go into some detail regarding work that suggest recall bias is increased with longer time periods and other factors, but there was no mention of studies that have shown no difference in reports when recall period varies, including 7- versus 30-day recall periods, and there are several such studies. in the end of the introduction, the authors state: "Previous reports comparing 30-day and 7-day recall periods on different self report measures have shown mixed results, with some measures showing no change in the distribution whereas others show a higher central tendency for 30-day relative to 7-day recall periods (4,14). again, no citations for studies that have found no difference. (e.g., K Flynn et al J Urol 2019; Lai et al, J Clin Epi 2009). 

Response: We acknowledge the lack of references to previous studies that found no effect of recall period and thank the reviewer for these suggested references. These have now been incorporated into the manuscript on pages 4- 5. 

Point 3. Did the authors examine whether there is any effect of order (first versus second) on responses? I appreciate the counterbalancing to control for order effects, but it remains of interest whether the second time around leads to more reporting (regardless of whether the second time around was 7-day or 30-day. 

Response: We have updated the analysis to account for a potential effect of time (first or second occasion of measurement) and for a potential interaction with recall period. No main effects of time or interaction effects between time and recall period were found. We have now reported this as our main models in the results (pages 9-11). 

Point 4. General: The approach to introduction and discussion generally reveals an experimenter mindset that there should be a difference where there was none. I mentioned this in intro, but then again in discussion are comments like: "Future research should investigate why certain emotional states might be more sensitive to changes in the recall period than others" and "caution should be taken in interpreting changes in individual items of the K6 scale as some items are more sensitive to the changed recall period than others". Perhaps a good balance would be to recommend more research to identify emotional (and other) states that are not sensitive to recall period and yet still valid." 

Response. We appreciate that the wording throughout the results and discussion section implied there should be significant results that were not found. We have updated the wording to more clearly indicate there was generally no effect of the recall period, and that future research could investigate which emotional or physical symptoms are unaffected or minimally affected by the recall period. We have made changes on pages 13 and 14.

Point 5. Minor: Intro: "The aim of the current study was to assess whether changing the recall period from 30 days to 7 days systemically alters ..." do you mean "systematically"?

Response: We thank the reviewer for catching this error and changed this to “systematically” on page 4. 

Point 6. Procedure: "After completing the K6 and TTPN item (with either the 7-day or 30-day recall period), participants completed an unrelated distractor task before completing the K6 and TTPN item again with the alternate recall period ..." Please provide the average amount of time the distractor task took.

Response: We have now provided more context and indicated that the total study took 10 minutes to complete on average, and that the interim task took approximately 7 minutes to complete in the Procedures section on page 8. 

Reviewer #2

An extremely clear and well-written piece of work, which answers an interesting question, which is relevant to many researchers who frequently use self-reported mental health scales. The authors investigated the effect that changing the recall period from seven to thirty days for the K6 and TTPN measures had on participant responses. There was a slight trend for higher scores for the thirty-day recall period, although this was not statistically significant and crucially did not change the interpretation of whether someone would be classified as having severe psychological distress. I was glad to see that they considered the effects of individual items from the K6 rather than just the overall sum score. 

Point 1. However, I was expecting to see a Table of results for the mixed effects models (the main analysis), could this be added for review prior to publication? 

Response. We have now added the results of the linear regression models in Table 3 on page 10. We have also more clearly indicated that the full results tables for the remainder of the analyses can be found at https://osf.io/aezt6/ (page 8). 

Reviewer #3:

This paper investigates a useful question for those interested in survey methodology related to mental health. Differences that might emerge depending on the requested recall period would have significant implications for future data collection and clinical considerations that rely on population surveys. This paper provides useful insights on the relative similarity among both 7-day and 30-day recall periods for self-assessed psychological distress.

Overall, I think the paper is well written, and both the methods and results are robust and accurately interpreted. The findings are straightforward and communicated well.

I do think, however, that more consideration could be given to the limitations of this study in the discussion section. The background section does a very good job highlighting some of the issues that might relate to differences in responses for alternate recall periods, and there is some good reflection of these issues in the discussion. Still, there are some specific limitations that I think deserve greater attention to strengthen the discussion and draw out the implications of the research:

Point 1. The methods section describes how the ORU sample was constructed, indicating that age and gender were collected to serve as covariates. I think the discussion should reflect on the limitations from not including other potentially important covariates, particularly those related to education, socioeconomic status, or other physical health measures. Clearly, the sample size restricts the complexity of the model, but I think the paper would benefit from more consideration of this idea. 

Response: We have expanded the limitations section of the paper to indicate that the sample was stratified by age and sex in an aim to be representative of the Australian population, we did not stratify by socio-economic or health related factors. This is included on page 13. 

Point 2. Relatedly, I think the paper should include some mention of the results related to the age and gender covariates. Perhaps there were no statistically significant results for these, but their inclusion in the models could also have indicated some kind of stratified differences, such that, for example, a difference between recall periods was identified for a specific age group. Such findings would inform future survey strategies. 

Response: We have now reported the full model results for the mixed regression analyses in a table as part of the main text, and the full outcomes of the analysis are provided in the data repository on the OSF. We note that the significant age effects shown in the main table reflect differences in mean distress scores between age groups, not an effect of the recall period. We have noted these effects in the Results section on page 10. Given that our study is not adequately powered to investigate age group effects and the lack of prior evidence suggesting that recall period effects might differ with age, we have opted not to investigate this further to avoid reporting potentially spurious findings. 

Point 3. The text might also benefit from considerations of the difference in the K6 and TTPN measures themselves, and what this might mean for other mental health measures used in population surveys. In other words, some surveys include CES-D for depression or GAD-7 for anxiety, which are intended to be more specific than general psychological distress. This would be for future research, but might be a point to briefly highlight in the limitations. 

Response: We have expanded our limitations section on pages 12-13 to suggest further investigation into symptom-specific inventories such as anxiety scales to determine whether certain symptoms are more sensitive to the recall period.

Point 4. The conclusions around the applicability of 7-day recall period responses to pre-pandemic 30-day recall measures are strong and supported well by the analysis presented. The authors might want to consider including some reflections or recommendations on the pros and cons of using 7-day vs 30-day measures in future survey settings.

Response: We have reiterated in the conclusion on pages 13-14 that since there is evidence that overall psychological distress scores are similar between the 7-day and 30-day recall period, that shorter recall periods should be used when sensitivity and accuracy are required, whereas longer recall periods should be used when the focus is on estimating the degree of persisting symptoms or the prevalence of severe distress. 

Point 5. There might also be value in considering the relevance of these findings for clinical settings related to mental health and psychology, given the logistic regression analyses covering likely clinical poor mental health. 

Response: We have added a sentence to our conclusion to reiterate that changing the recall period is unlikely to alter a person’s clinical status on page 14.

---

## [Decision Letter · Decision Letter 1]

23 Nov 2023

Comparing estimates of psychological distress using 7-day and 30-day recall periods: Does it make a difference?

PONE-D-23-09923R1

Dear Dr. Chilver,

We’re pleased to inform you that your manuscript has been judged scientifically suitable for publication and will be formally accepted for publication once it meets all outstanding technical requirements.

Kind regards,

Qin Xiang Ng, MBBS, GDMH, MPH

Academic Editor

PLOS ONE

Additional Editor Comments (optional):

Reviewers' comments:

Reviewer's Responses to Questions

**Comments to the Author**

1. If the authors have adequately addressed your comments raised in a previous round of review and you feel that this manuscript is now acceptable for publication, you may indicate that here to bypass the “Comments to the Author” section, enter your conflict of interest statement in the “Confidential to Editor” section, and submit your "Accept" recommendation.

Reviewer #2: All comments have been addressed

Reviewer #3: All comments have been addressed

2. Is the manuscript technically sound, and do the data support the conclusions?

Reviewer #2: Yes

Reviewer #3: (No Response)

3. Has the statistical analysis been performed appropriately and rigorously? 

Reviewer #2: Yes

Reviewer #3: (No Response)

4. Have the authors made all data underlying the findings in their manuscript fully available?

Reviewer #2: Yes

Reviewer #3: (No Response)

5. Is the manuscript presented in an intelligible fashion and written in standard English?

Reviewer #2: Yes

Reviewer #3: (No Response)

6. Review Comments to the Author

Reviewer #2: The paper is clear and well written and the authors have done a good job at addressing all of the reviewers comments.

Reviewer #3: (No Response)

7. PLOS authors have the option to publish the peer review history of their article (what does this mean?). If published, this will include your full peer review and any attached files.

Reviewer #2: No

Reviewer #3: **Yes: **Dr Brian Beach

---

## [Editor Report · Acceptance letter]

30 Nov 2023

PONE-D-23-09923R1 

Comparing estimates of psychological distress using 7-day and 30-day recall periods: does it make a difference? 

Dear Dr. Chilver:

I'm pleased to inform you that your manuscript has been deemed suitable for publication in PLOS ONE. Congratulations! Your manuscript is now with our production department. 

Kind regards, 

on behalf of

Dr. Qin Xiang Ng 

Academic Editor

PLOS ONE